# UV Light Causes Structural Changes in Microplastics Exposed in Bio-Solids

**DOI:** 10.3390/polym15214322

**Published:** 2023-11-04

**Authors:** Somayye Sadat Alavian Petroody, Seyed Hossein Hashemi, Luka Škrlep, Branka Mušič, Cornelis A. M. van Gestel, Andrijana Sever Škapin

**Affiliations:** 1Environmental Sciences Research Institute, Shahid Beheshti University, Tehran 1983963113, Iran; maede.alavian@gmail.com (S.S.A.P.); h_hashemi@sbu.ac.ir (S.H.H.); 2Slovenian National Building and Civil Engineering Institute, Dimičeva ulica 12, 1000 Ljubljana, Slovenia; luka.skrlep@zag.si (L.Š.); branka.music@zag.si (B.M.); 3Institute for Life and Environment (A-LIFE), Faculty of Science, Vrije Universiteit Amsterdam, De Boelelaan 1085, 1081 HV Amsterdam, The Netherlands; kees.van.gestel@vu.nl; 4Faculty of Polymer Technology—FTPO, Ozare 19, 2380 Slovenj Gradec, Slovenia

**Keywords:** microplastics, degradation, hydroxyl, carbonyl, ultraviolet light

## Abstract

Bio-solids (biological sludge) from wastewater treatment plants are a significant source of the emission of microplastics (MPs) into the environment. Weakening the structure of MPs before they enter the environment may accelerate their degradation and reduce the environmental exposure time. Therefore, we studied the effect of UV-A and UV-C, applied at 70 °C, on three types of MPs, polypropylene (PP), polyethylene (PE), and polyethylene terephthalate (PET), that are commonly found in sewage sludge, using three shapes (fibers, lines, granules). The MPs were exposed to UV radiation in bio-solid suspensions, and to air and water as control. The structural changes in and degradation of the MPs were investigated using Attenuated Total Reflectance–Fourier Transform Infrared Spectrometry (ATR-FTIR) and surface morphology was performed with SEM analysis. UV exposure led to the emergence of carbonyl and hydroxyl groups in all of the PP samples. In PE and PET, these groups were formed only in the bio-solid suspensions. The presence of carbonyl and hydroxyl groups increased with an increasing exposure time. Overall, UV radiation had the greatest impact on the MPs in the bio-solids suspension. Due to the surface-to-volume ratio of the tested samples, which influences the degradation rate, the fibers were more degraded than the other two plastic shapes. UV-A was slightly more effective at degrading the MPs than UV-C. These findings show that ultraviolet radiation in combination with an elevated temperature affects the structure of polymers in wastewater bio-solids, which can accelerate their degradation.

## 1. Introduction

Global plastics production was estimated at more than at 390 million metric tons in 2022 [1]. In a recent paper by Lim, it is estimated that almost 400 million tons of plastics are produced each year and that this amount may double by 2050 [2].

While the benefits of using plastic are high [3,4], this valuable commodity has raised environmental concerns that it enters different environments through improper disposal [5,6]. The disposal of plastic waste is difficult due to its durability and high resistance against degradation [7,8]. While some plastic wastes are recycled, most end up in landfills [7,9].

Despite the widespread study of the harmful effects of plastics on the environment, a major concern in recent years has been the presence of smaller pieces of plastic, or microplastics (MPs), which are plastics smaller than 5 mm in all shapes [10]. This small plastics debris has been identified in various aquatic and terrestrial environments and has a high stability and long environmental residence times [11], which has raised environmental concerns [12,13,14,15]. They also have a high potential for sorbing and transporting pollutants [11,16,17] and may cause physical and chemical damage to organisms and, ultimately, humans [18,19,20,21].

Wastewater treatment plants (WWTPs) play an important role in releasing MPs into aqueous and soil environments through effluents and bio-solids (biological sludge) [10,22,23]. Studies on bio-solids as an important source of MPs in the environment indicate that a large number of MPs enter the environment daily [10,14]. For example, Alavian Petroody et al. [10] estimated that more than 423 MPs/m^3^ and 129 MPs/g of dry weight enter the environment through wastewater and bio-solids from the Sari treatment plant in Northern Iran, respectively, the majority of which are fibres with a size smaller than 500 µm.

Therefore, it may be very effective and necessary to develop solutions to remove [24] or weaken these polymers before their disposal or the land application of bio-solids [11]. Laboratory studies have investigated the effect of UV [25,26] and also of other factors such as mechanical abrasion in simulated natural conditions on the stability of MPs under sunlight in the sea or on land [27,28]. Rare studies have aimed to use the factors affecting degradation as a positive factor in weakening MPs in an environment such as bio-solids [29]. Due to their high nutrient content, bio-solids produced in wastewater treatment plants may be used as fertilizer or as a soil modifier in agriculture and forestry. However, the presence of high amounts of MPs in bio-solids has attracted the attention of many researchers [30,31,32]. The average of 129 microplastics per gram of dry weight in the dewatered sludge of a treatment plant in Iran reported by [33] corresponds with a daily environmental emission of about 276 million microplastics. Studies in other countries have shown the presence of from 0.370 to 101 microplastics per gram of sludge in the United States and Europe [34,35,36,37,38,39]. Therefore, weakening MPs in bio-solids before their release into the environment can be considered as a possible option to accelerate their environmental degradation.

Only limited information, however, is available on the influence of the defined polymer type and its different shapes on MP stability following exposure to various UV light under different conditions. In this study, the effect of UV-A and UV-C on the degradation of MPs was determined in wastewater bio-solids. We hypothesized that UV exposure and the presence of organic matter or impurities would have an important effect on the formation of hydroxyl and carbonyl groups on the microplastics in bio-solids. The effects of UV-A and UV-C were determined at 70 °C, which is the temperature used to reduce pathogens in bio-solids. Three types of widely used polymers, polyethylene (PE), polypropylene (PP), and polyethylene terephthalate (PET), which are highly abundant in the bio-solids of WWTPs, were exposed in three different shapes (fibres, lines, granules). The final aim was to find conditions which enhance the degradation of MPs and consequently contribute to reducing their emission through wastewater bio-solids and their longevity in the environment.

## 2. Materials and Methods

### 2.1. Synthetic Wastewater Bio-Solids

To investigate the effect of UV at an elevated temperature on the structure of MPs, bio-solid suspensions without any MPs were prepared in the laboratory in an aerated bioreactor which was fed with water, nutrients, and microorganisms, as well as fed with human feces. Air was supplied by an air pump and fine bubble diffusers. The mixed liquor-suspended solids (MLSS) concentration of the synthetic bio-solids suspension was maintained at around 7000 mg/L in all tests to be similar to the MLSS of the secondary bio-solids concentration in conventional wastewater treatment plants.

### 2.2. UV and Temperature Exposures

A test chamber (120 cm × 30 cm × 50 cm for length, width, and height, respectively) with shiny (mirror type) steel plates at its inner body was used. It was equipped with five UV lamp fittings, nine magnetic heater stirrers, temperature and UV intensity sensors, and a control and recording unit. The temperature was controlled online by the sensors under the sample containers as well multi-point measurements inside the chamber. Apart from this, periodically, sample temperature was measured with the help of thermometers. The high flux of UV light caused all the UV records to be out of range of the sensor. Thus, the UV radiation flux was calculated and reported based on the power of the lamp. The area of the chamber was 0.36 m^2^ (120 cm × 30 cm). Exposure times of 24 h and 48 h were applied. Five UV-A (340 nm, Philips, Amsterdam, The Netherlands) or 5 UV-C lamps (254 nm, Mazda, GE, USA) (40 W per lamp) were used separately in the different experiments, positioned approximately 30 cm above the water and bio-solid samples, and approx. 35 cm above the air samples—see Figure 1 for schematic of setup.

Three types of polymers (polyethylene (PE), polypropylene (PP), and polyethylene terephthalate (PET)) were tested in three different shapes: fibres, lines, and granules. Granules can also be called pellets due to their cylindrical shape in a size of about 3 mm in diameter and about 5 mm in height—see Appendix A. The thickness of fibres and lines was about 50 µm and 500 µm, respectively, while the length of lines was approximately 15 mm. Both were obtained from consumer factories. The samples contained no polymer additives which may have affected the peaks observed in the FTIR analysis. Grades of PP, PE, and PET were Z30S, 0035, and BG825/TG 641, respectively. All materials were made in Teheran, Iran: in the case of PP for use in factories for production of rope, dress, and medical applications, in the case of PE for production of pipes, tanks, and toys, and in the case of PET for production of bottles, yarn, and dress. In the case of PP and PET, all three shapes were tested, while for PE only the lines and granules were used because no PE fibres could be obtained.

The samples were tested in three environments (bio-solid suspensions, water, and air) to investigate the effects of the two types of UV light at exposure times of 24 h and 48 h on MP degradation. Exposures were performed at 70 °C because it has been mentioned as a suitable temperature for polymer destruction [40]. Both water and air environments were used as controls to determine the role of bio-solids along with UV in causing changes in the polymers.

Polymer samples (700 fibres, 30 lines, and 30 granules for each test) were exposed in 600 mL glass beakers (height: 12 cm, diameter: 10.5 cm). For testing in water and bio-solids, a volume of 300 mL of water or 300 mL of bio-solids suspension was used for each beaker. The volume of water and bio-solids suspension was kept constant during the test by frequently adding additional water to compensate for evaporation. Bio-solid suspensions and water samples were continuously stirred using a magnetic stirrer (approx. 100 rpm) to ensure homogenous exposure to the UV light.

### 2.3. Extraction of Samples

MPs from the exposed water samples were extracted by passing the water through a stainless steel filter and a cellulose filter (Whatman, Ashless, No. 42, 125 mm, Cytiva, Maidstone, UK) was placed under it to ensure that the samples did not come out. The samples taken for testing were from stainless steel filters. Then, the samples were washed with distilled water. Also, the MPs from the bio-solid suspensions were extracted by washing with distilled water after passing through a filter and were separated using a microscope. After separating the MPs from the bio-solids, they were washed several times with distilled water to remove any adhering residual matters. We checked by microscopy that no biological contamination was placed on the samples. All MPs extracted from the water and bio-solid samples were dried in an oven at 60 °C for 24 h and, together with the MPs extracted from the air environment, stored in glass vials in the dark at room temperature [41].

### 2.4. FTIR and SEM Analysis

The possible changes in the surface of the investigated polymers particles were examined using a scanning electron microscope (Model XL30, Philips, The Netherlands). Accelerating voltage was set to 25 kV. Images were obtained with detection of secondary electrons.

To investigate the structural changes caused by UV exposure, Fourier Transform Infrared Spectrometry (FTIR)-Attenuated Total Reflectance (ATR; diamond crystal) was applied using a FTIR spectrometer (Spectrum Two, Perkin Elmer, Waltham, MA, USA). Spectra were recorded in the range of 400–4000 cm^−1^ with an average of 4 scans at 4 cm^−1^ resolution. A thin part of the granules was cut for the analysis. The outer surface was applied to the ATR crystal. Fibre and line samples were analysed without additional preparation. Baselines of FTIR spectra were corrected by the baseline correction function in the program Spectrum IR 10.6.1. Then, the spectrums were normalized to identify changes in the structure of the polymers, and functional groups indicative of weathering (carbonyl and hydroxyl groups) were identified. For each MP type, the FTIR-ATR spectra before and after exposure to the different UV types in the different media were compared, and changes in the presence and abundance of functional groups were examined.

To check for the repeatability of the FTIR analyses, the measurements of a few random samples were repeated several times. The spectra were practically the same, confirming high repeatability. Therefore, only one spectrum was recorded for each sample.

### 2.5. Indexes of Hydroxyl and Carbonyl Group Bonds

To compare the intensity of hydroxyl and carbonyl groups formed in the MPs exposed in the bio-solid suspensions with other studies in the natural environment and to assess the importance and effect of UV-A and UV-C in the formation of hydroxyl and carbonyl groups, their bond indices were used. These indices were calculated following [41]:(1)$$Hydroxyl\;Index\;\left(HI\right)=\frac{the\;maximum\;peak\;absorbance\;for\;hydroxyl\;group}{the\;value\;of\;a\;reference\;peak}$$

**Note** **1.**Maximum peak Absorbance for Hydroxyl (PP, PE, PET): 3000–3500 cm^−1^. Reference peak: PP = 2800–3000 cm^−1^, PE = 2900–3000 cm^−1^, PET = 1600–1800 cm^−1^.


(2)
$$Carbonyl\;Index\;\left(CI\right)=\frac{the\;maximum\;peak\;absorbance\;for\;Carbonyl\;group}{the\;value\;of\;a\;reference\;peak}$$


**Note** **2.**Maximum peak Absorbance for Carbonyl (PP, PE): 1550–1800 cm^−1^. Reference peak: PP = 2800–3000 cm^−1^, PE = 2900–3000 cm^−1^.

Also, the effects of UV-A and UV-C on the structural groups created in each sample were compared by the relative destructive power of the UV-A to the UV-C index (RDP_A,C_):(3)$${RDPH}_{A,C}=\frac{{HI}_{UVA}}{{HI}_{UVC}}$$
(4)$${RDPC}_{A,C}=\frac{{CI}_{UVA}}{{CI}_{UVC}}$$

**Note** **3.**HI_UVA_ = the HI for sample in the bio-solid suspension after t hours exposure to UV-A. HI_UVC_ = the HI for sample in the bio-solid suspension after t hours exposure to UV-C. CI_UVA_ = the CI for sample in the bio-solid suspension after t hours exposure to UV-A. CI_UVC_ = the CI for sample in the bio-solid suspension after t hours exposure to UV-C. t = the exposure time. 

## 3. Results

### 3.1. Effect of UV on PP

PP samples exposed to UV-A and UV-C at 70 °C in air, water, and bio-solids showed the emergence of carbonyl (C=O, ~1700 cm^−1^) and hydroxyl (OH, ~3300 cm^−1^) groups (Figure 2)—see Appendix A for 400–4000 cm^−1^ wavenumber region. The PP samples exposed in the air environment (first control) showed an increasing formation of C=O and OH groups with an increasing exposure time. The results also showed an effect of the physical shape on the formation of these groups, with the fibres showing more of these groups than the lines, followed by the granules. In general, the intensity of the formed groups was low in the line and granule samples.

The PP samples exposed to UV in the water environment (second control) also showed an increased abundance of hydroxyl and carbonyl groups with an increased time of exposure (Figure 2). Also, in water, the intensity of the groups that formed was higher in the fibres, followed by the lines, and was lowest in the granules. However, slightly more hydroxyl and carbonyl groups were formed in the water than in the air environment.

Changes in the PP samples exposed to UV light in the bio-solids suspension indicate the emergence of oxidation groups, which were also observed in the control environments (air and water): Figure 2. In this case, the intensity of the oxidation group formation also increased with time, and was higher in the fibres, followed by the lines and granules. However, the intensity of the formed groups in the PP samples was much higher in the bio-solid suspensions than in the air and water environments (Figure 2).

UV-A exposure was slightly more effective in forming carbonyl and hydroxyl groups in the polypropylene samples than UV-C (Table 1).

SEM images of the surfaces of PP lines after 48 h exposure to UV-A, PP granules after 24 h exposure to UV-A, and PP granules after 48 h exposure to UV-C are presented in Appendix A, respectively. In contrast to FTIR spectra which presented major chemical degradation of the polymer, the SEM images presented minor morphological changes.

Several researchers have mentioned that physical changes are often observed in long time tests and, usually, there is no clear correlation between the occurrence of chemical degradation and physical changes over a short UV exposure time. For example, some researchers [42,43] observed minor morphological changes for most of their aging treatments, which was in contrast with the results gained from ATR-FTIR and CI. They stated that, in order to investigate the physical changes in polymers (based on SEM and XRD analysis), an exposure time of 96 h or less does not cause much change in the physical structure of the polymer, while significant chemical changes may have occurred in the polymer.

### 3.2. Effect of UV Light on PE Samples

The exposure of PE samples to UV light in the control water and air environments did not induce any changes in their structure, so no formation of carbonyl and hydroxyl groups was observed. The presence of a small number of hydroxyl and carbonyl groups in the FTIR spectra was visible due to their presence in the main structure of the pristine PE samples (Figure 3)—see Appendix A for 400–4000 cm^−1^ wavenumber region.

The irradiation of PE samples with UV-A and UV-C light in bio-solid suspensions produced hydroxyl groups only in lines but not in granules, and their intensity increased with time. It was difficult to assess the formation of carbonyl groups in the PE due to their presence in virgin samples. Nevertheless, a significantly increased intensity of carbonyl groups was observed in the PE lines after 48 h of UV-A exposure (Figure 3). UV-A light had a greater effect than UV-C light on the formation of functional groups in the PE line samples in the bio-solid suspensions (Table 2).

SEM images of the surface of PE lines after 48 h exposure to UV-A, PE granules after 24 h exposure to UV-A, and PE granules after 48 h exposure to UV-C are presented in Appendix A, respectively. The same as for the PP samples, the SEM images present minor morphological changes.

### 3.3. Effect of UV Light on PET Samples

The exposure of different PET samples to different exposure conditions only produced changes in the fibers and lines in the bio-solids, leading to an increased abundance of hydroxyl groups, which increased with exposure time. No changes were observed in the other samples (Figure 3)—see Appendix A for 400–4000 cm^−1^ wavenumber region.

Also, UV-A exposure had a slightly stronger effect than UV-C light on the formation of functional groups in the PET fiber and line samples in the bio-solid suspensions (Table 3).

The specific peaks used as a reference and the exact calculation may vary depending on the material used (sample source, age, storage, preparation, and other effects on the sample). For this reason, the CI and HI were calculated for the virgin samples (the samples at 0 min of UV irradiation). These values are around 1 and are presented in Appendix A.

The possible pathway of degradation of PE and PP polymers is different from that of PET polymers; these are adopted from [44] and are shown in Figure 4 and Figure 5.

## 4. Discussion

### 4.1. Photodegradation of Polymers

The released plastic waste is affected by abiotic and biotic environmental factors, which is referred to as weathering and causes a loss of the physical integrity of the material [45]. Most of the degradation of plastics occurs in the topsoil due to direct exposure to UV radiation, increased oxygen availability, moist conditions, and higher temperatures [11,46]. In general, the degree of plastic degradation depends on external conditions [25,47].

A wide range of plastics released into the environment absorb ultraviolet (UV) radiation and undergo photolytic, photo-oxidation, and thermal-oxidation reactions that lead to their degradation [48,49]. The photodegradation of a polymer involves physical and chemical changes, which lead to a reduction in their molecular weight and the loss of their flexibility and mechanical integrity [50,51]. Photodegradation may occur in the presence of oxygen (photo-oxidation) [48]. The photo-oxidation of polymers typically involves reactions caused by free radicals that lead to chain cleavage, cross-linking, and eventually the formation of functional groups [52,53]. For this purpose, light must be absorbed by the polymer. Therefore, the presence of chromophoric groups in macromolecules is a prerequisite for the initiation of any photochemical reaction. In general, the factors that cause the photodegradation of polymeric materials can be divided into two categories [52]:Internal impurities (i.e., hydroperoxide, carbonyl, unsaturated bonds, catalyst residues);External impurities, including chromophoric groups, solvents, catalysts, additives, and metals and metal oxides.

The absorption of light by these chromophores leads to the formation of radicals. When free radicals are formed along polymer chains, they undergo more reactions, eventually leading to the production of alcohols, acids, and aldehydes [54,55,56,57,58]. The formation of hydroxyl products (e.g., hydroperoxide, alcohol, and carboxylic acid) (3200–3600 cm^−1^), carbonyl (e.g., ketones, esters, and carboxylic acids) (1700–1800 cm^−1^), and unsaturated groups (e.g., Vinylidene) (800–1000 cm^−1^) has been observed in studies on the exposure of polymers to natural weathering conditions [59].

Different results were observed for the effects of UV-A and UV-C light on the degradation of PE, PP, and PET in different shapes, representing the predominant types of MPs in WWTPs [10]. All forms of PP were degraded by UV-A and UV-C light, but for PE, only the line shape was degraded in the bio-solid suspension (PE fiber was not studied). Also, slight changes in the PET fibers and very small changes in the line shapes were observed in the bio-solids suspension under UV irradiation. The finding that polyolefins (PP and PE) are less resistant to degradation than PET are in line with the results in the literature. The difference can be attributed to the lower molecular weight of polyolefins compared to PET [26]. In fact, PP is much more sensitive to photodegradation than PE [60].

In our research, it seems that the main degradation mechanism of polymer samples in air is the collision of UV light photons with the polymer, while, for the polymer samples in water, it is degradation by the collision of UV light photons with the polymer along with the degradation of polymer by free radicals formed in water; and for the bio-solid, the most important mechanism is the formation of free radicals in the rich organic matter liquid and the degradation of the polymer by them. These mechanisms can lead to the degradation of polymers at different rates.

Albertsson et al. [61] showed that there is a synergetic interaction between photo-oxidation and biodegradation. More biodegradation occurs when hydroxyl and carbonyl groups, which are signs of weathering and photodegradation, are formed in plastic samples [62]. The photo-oxidation of polyolefin changes its highly hydrophobic surface area to become less hydrophobic, decreases its molecular weight, and decreases its tensile strength; thus, it favors biodegradation [26]. Therefore, before investigating the effects of biodegradation on plastics, some researchers exposed them to UV radiation to create some of these functional groups [62]. As a consequence, it can be stated that the formation of oxidation groups in polyolefin samples in the sludge environment can help their biodegradation after entering the environment.

### 4.2. Structural Changes of PP

#### 4.2.1. Impact of Environment

The changes in PP due to UV exposure at 70 °C were the most pronounced in the bio-solids suspension, producing the most hydroxyl and carbonyl functional groups, which are indicators of the degradation of the PP structure. The simultaneous action of high temperatures and UV radiation increases the possibility of initiating destructive reactions, with the main reaction being based on the generated free radicals [40,52]. Therefore, the high degradation of polypropylene in the bio-solids suspension can be due to the presence of organic matter along with water. This is confirmed by Da Costa et al. [63], who found that the presence of organic matter in the environment triggered the creation of hydroxyl and carbonyl groups in the structures of plastics. The highly abundant organic matter in the bio-solids can be degraded by UV light, stimulated by the high temperature, leading to the formation of radicals [40,52,64]. Along with hydroxyl radicals (•OH) formed due to the high moisture level of the bio-solids, these environmental conditions lead to an increased degradation rate of polypropylene [64]. It is also possible that, due to the high turbidity of the bio-solids suspension, more light was absorbed than in the other environments (water, air), which also increased the formation of destructive polymer radicals and caused more damage to the microplastics, due to increasing the temperature of the turbid water [65].

The formation of hydroxyl and carbonyl groups in the PP samples was higher in water than in air, but lower than in the bio-solids. It can therefore be hypothesized that the formation of hydroxyl and carbonyl groups in PP samples in the water environment was due to the presence of high humidity and oxygen in the test chamber. Brandon et al. [41] also stated that the destruction of microplastics in seawater was greater than in an air environment, which can be caused by destructive agents such as hydroxyl radicals. However, even without organic matter and moist conditions, the PP samples exposed to ultraviolet radiation in air did show some signs of damage, which indicates the impact of the combined heat–photo-oxidation degradation process on the polypropylene.

Studies on the effect of UV radiation and temperature on polypropylene have shown that heat treatment at 70 °C caused faster disintegration of the polymer chain [40,66]. Therefore, with increasing temperature, under the same radiation conditions, more functional groups representing oxidation products are produced [65,67]. At high temperatures, moisture, solvents, plasticizers, and volatiles are typically lost. Polypropylene and its copolymers become very brittle at high temperatures due to the loss of plasticizers [40]. Their molecular weight loss depends on the temperature and duration of the heat treatment. Also, the rate of the formation of volatile products is a function of the temperature. In addition, the main effect of increasing the temperature in the degradation process is related to increasing the mobility of radicals for the reaction. Therefore, high temperatures accelerate the degradation process by enhancing the formation of carbonyl and hydroxyl groups [25,40,68]. It may be concluded that the presence of impurities and moisture, as well as the high temperature, enhanced the effectiveness of the UV radiation in degrading PP.

#### 4.2.2. Impact of Physical Shape

The PP fibers showed the highest UV degradation, followed by the lines and granules. PP fibers are generally produced from PP granules, and the production of different types of fibers requires different production methods as well as the use of different metal catalysts [64]. The physical structure of fibers as well as other products produced from granules, such as lines, depends on the process of converting granules into products and factors which affect production [64,69]. The application of a higher temperature and mechanical processes on the molten granules increases the formation of impurities in manufactured products, which may enhance their degradation following exposure to the destructive effect of UV light [69]. As a consequence, we observed more degradation (formation of oxidation groups) in the fibrous and linear samples compared to the granules after UV exposure.

Ahmadi et al. [70], investigating the effect of photo-oxidation on PP lines and films, stated that the amount of internal chromophore groups initiating photodegradation, such as hydroperoxide groups, is higher in lines than in films. Therefore, the fibers and lines made of granules are more sensitive to photodegradation compared to granules.

UV degradation is a surface process, and the degradation of polymers starts from the surface and eventually reaches deeper layers [58,71]. The extent and intensity of polymer degradation depends on its thickness [40]. Therefore, the lower thickness of PP polymers leads to a higher surface-to-volume ratio and a greater impact of degradation processes [40]. Thus, according to the results obtained from this study, fibers, having the smallest thickness, suffered the highest degradation followed by the lines and, finally, the granules.

### 4.3. Influence of UV Radiation on Structural Changes of PE and PET

The results confirmed that ultraviolet irradiation affects the aging process of the tested polymers, polyethylene (PE), polyethylene terephthalate (PET), and polypropylene (PP) [72]. A greater impact of UV light on PE and PET was observed, with PE aging the easiest under artificial solar radiation [73]. The formation of carbonyl and hydroxyl groups in these polymers was observed only in the bio-solids suspension, which, as mentioned earlier, can be attributed to the presence of organic matter as well as moisture, and their role in causing structural changes in these polymers. Also for PE, the importance of thickness in creating oxidation groups was shown, with carbonyl and hydroxyl groups observed only in lines but not in granules.

Also, for PET samples, UV exposure only affected the line and fiber samples in the bio-solid suspensions, leading to an increased abundance of hydroxyl groups, with the fibers being the most affected. This could, again, be due to their lower thickness and higher surface-to-volume ratio.

### 4.4. The Effect of UV Type on Polymer Degradation

UV-A and UV-C exposure in different environmental conditions had almost the same effect on the structural changes in polymers, with the effect of UV-A being slightly greater than that of UV-C. The ratios of hydroxyl and carbonyl indices of PP samples in bio-solid suspensions following UV-A and UV-C irradiation ranged between 1.0 and 1.8. This indicates the relative importance of UV-A compared to UV-C. This index was from 1.4 to 4.1 and ~1.0 to 1.9 for PE and PET, respectively (Table 1, Table 2 and Table 3).

### 4.5. Comparison of Structural Changes in Samples in Bio-Solids with Other Studies

It was observed that functional groups in polymer samples in the bio-solids suspension under the influence of UV, especially UV-A after 48 h, cause their weakening and structural destruction before entering the environment. Therefore, it is possible that, after entering the environment through the bio-solids treated with UV and high temperature (70 °C), they are destroyed faster than MPs that enter the environment without treatment. Few studies have been performed on changes in MPs under natural conditions. Brandon et al. [41] showed that slight structural changes occurred in PE and PP MPs after 3 years and only small amounts of hydroxyl and carbonyl groups were formed, indicating their persistence in the environment [11]. The index of hydroxyl and carbonyl groups created in the microplastics samples in the bio-solids suspension of our experiment was higher than the indices calculated by Brandon et al. [41] under the influence of ambient conditions (Table 4). This indicates that MPs can be weakened by the combined effect of UV radiation and a high temperature in a short period of time (48 h) before entering the environment. As a result, they can be destroyed much faster after entering the environment.

The emergence of hydroxyl and carbonyl functional groups reduces the hydrophobicity of MPs and increases their sensitivity to biodegradation [62]. Based on this, it can be hypothesized that the degraded MPs are more suitable for microbial degradation. Bacteria may consume functional groups such as carbonyl, facilitating the fast degradation of MPs in the environment [26].

The degradation of plastics such as polyolefins in nature is very slow and first starts by abiotic environmental factors followed by microorganisms [72]. Also, the lack of active functional groups makes them resistant to biodegradation. Therefore, to make polyolefins degradable in the environment, it is necessary to decrease their hydrophobicity by oxidation. This may be achieved by treatments such as UV, heat, and chemicals, leading to the formation of carbonyl and hydroxyl functional groups [26]. Wilkes and Aristilde [73] stated that the degradation of plastic polymers without pretreatment with UV took longer, while the emergence of functional groups in polymers by pretreatment facilitated their degradation by bacteria [62].

Research on wastewater treatment plants has shown that most MPs entering bio-solids have a size of less than 500 µm [10,14], meaning a very high surface-to-volume ratio and very low thickness. According to the results of this study, the lower the thickness of the polymers, the greater the impact of factors such as UV light on their structural degradation. Therefore, it is likely that the application of UV irradiation in wastewater treatment plants will have a favorable effect on MPs because of their small size. Using this method before the bio-solids dewatering process not only may eliminate pathogens but also weakens the MPs and accelerates their microbial degradation after entering the environment. In recent years, there have been many concerns about the degradation of microplastics due to environmental factors such as UV and the formation of nano plastics and their negative effects on the environment. Of course, it should be noted that becoming smaller particles, i.e., in the nanometer size range, does not necessarily mean degradation but may also be due to fragmentation. The degradation of larger sizes by creating functional groups, such as hydroxyl and carbonyl groups, however, does indicate degradation. Therefore, it can be stated that, if nano plastics are formed in the experimental environment under the influence of UV, these particles have hydroxyl and carbonyl groups, unlike nano plastic particles that may be formed due to erosion and wear in the environment that lack these functional groups and the formation of which in plastics in nature may take several years [41]. The importance of creating these groups in these very small particles is that they are destroyed faster by biological agents after entering the environment. The introduction of functional groups reduces the hydrophobicity of polymers and provides the conditions for biological agents to adhere and cause degradation [26,62].

Further, deeper research is underway to use this method as an effective polymer destruction technique in sewage treatment plants.

## 5. Conclusions

The entry of MPs in bio-solids into agricultural lands is an important environmental issue. Weakening MPs before releasing them into the environment will lead to faster destruction. For this purpose, for the first time, the effect of destructive processes on the weakening of MPs in bio-solids was studied. This study determined the effects of two types of UV radiation on three types of polymers commonly found as MPs, tested in three different shapes (fibers, lines, granules), at a temperature of 70 °C, which is commonly applied to kill microbes in wastewater bio-solids. The UV exposure of the MPs in the bio-solids had the greatest effect on the formation of hydroxyl and carbonyl groups compared to control water or air environments. The presence of organic matter or impurities in the bio-solids and their moisture content probably had an important effect on the formation of hydroxyl and carbonyl groups on the microplastic samples. Polypropylene (PP) was the most susceptible to irradiation effects in different environments followed by polyethylene (PE) and polyethylene terephthalate (PET). Due to their having the highest surface-to-volume ratio, the fibers showed the highest degradation, followed by the lines and granules. UV-A light seems slightly more effective in affecting the MPs than UV-C light, although the differences were small. It may be concluded that exposing MPs in wastewater bio-solids to UV light at an elevated temperature of 70 °C may be effective in weakening their structure and potentially reducing their lifetime in the receiving environment.

## Figures and Tables

**Figure 1 polymers-15-04322-f001:**
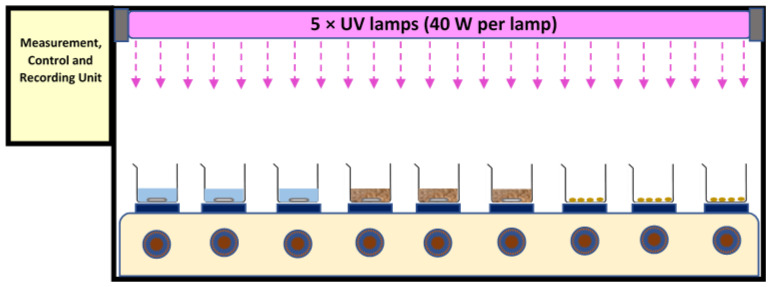
Schematic of setup for UV aging tests, including a test chamber (L: 120 cm × W: 30 cm × H: 50 cm), 5 × UV lamps, 9 × heater stirrers, and a control unit.

**Figure 2 polymers-15-04322-f002:**
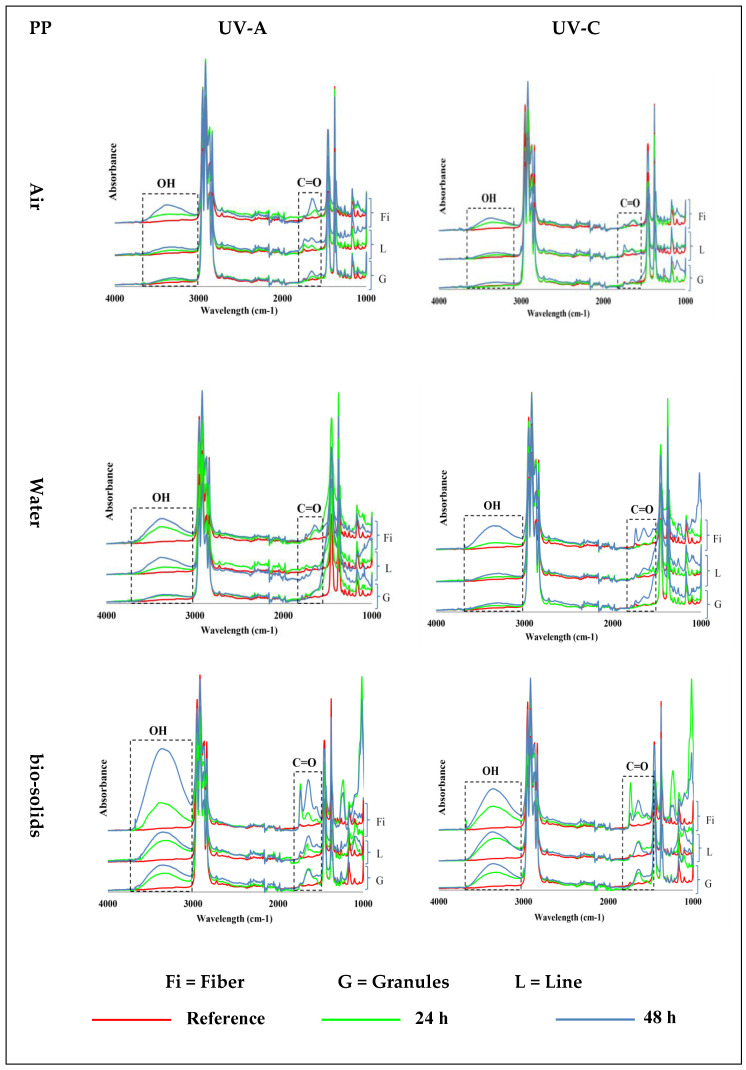
Induction of carbonyl (C=O; ~1700 cm^−1^) and hydroxyl (OH, ~3300 cm^−1^) groups in different shapes of polypropylene (PP) microplastics in air (**top**), water (**middle**), and bio-solid suspensions (**bottom**) exposed for 24 h or 48 h to UV-A (**left**) and UV-C (**right**) light, at 70 °C, and compared with virgin (reference) samples. (Y axes are related to the adsorption peaks.).

**Figure 3 polymers-15-04322-f003:**
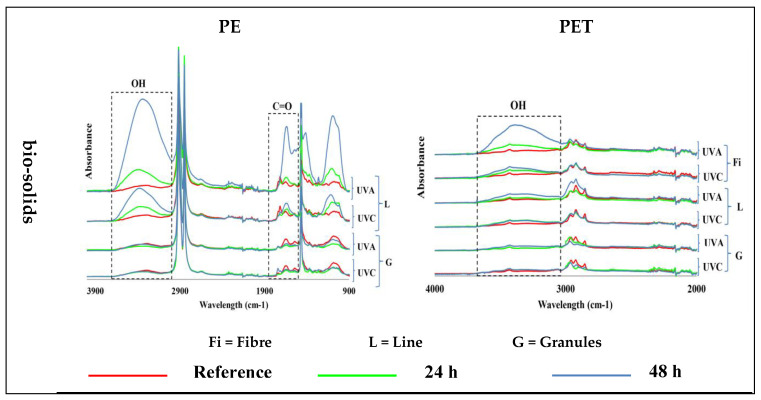
Induction of carbonyl (C=O; ~1700 cm^−1^) and hydroxyl (OH, ~3300 cm^−1^) groups in different shapes of polyethylene (PE; **left**) and polyethylene terephthalate (PET; **right**) microplastics exposed for 24 h and 48 h to UV-A and UV-C light, at 70 °C, in different environments at different times, and compared with virgin (reference) samples. (Y axes are related to the adsorption peaks.).

**Figure 4 polymers-15-04322-f004:**
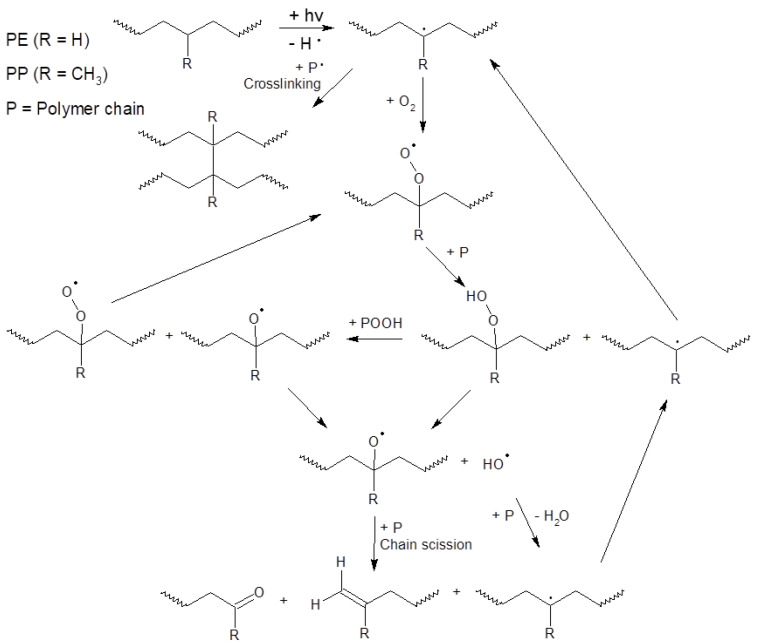
Scheme of photo-initiated degradation pathway of PE and PP polymers adopted from reference [44].

**Figure 5 polymers-15-04322-f005:**
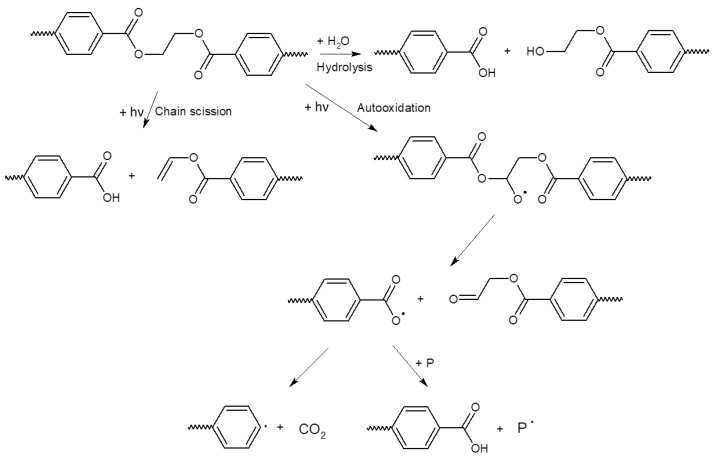
Scheme of hydrolytic degradation and photo-initiated degradation pathway of a PET polymer adopted from reference [44].

**Table 1 polymers-15-04322-t001:** Comparison of the effect of UV-A and UV-C on the formation of carbonyl (C=O) and hydroxyl (OH) groups in PP samples in the bio-solid environment (${\mathrm{RDPH}}_{A,C}=\frac{{\mathrm{HI}}_{\mathrm{UVA}}}{{\mathrm{HI}}_{\mathrm{UVC}}}\;\mathrm{and}\;{\mathrm{RDPC}}_{A,C}=\frac{{\mathrm{CI}}_{\mathrm{UVA}}}{{\mathrm{CI}}_{\mathrm{UVC}}}$).

RDP_A,C_	Fibre		Line		Granule	
	OH	C=O	OH	C=O	OH	C=O
24 h	1.5	1.3	1.2	1.0	1.2	1.3
48 h	1.8	1.5	1.3	1.8	1.0	1.3

**Table 2 polymers-15-04322-t002:** Comparison of the effect of UV-A and UV-C on the formation of carbonyl (C=O) and hydroxyl (OH) groups in PE line samples in the bio-solids environment (${\mathrm{RDPH}}_{A,C}=\frac{{\mathrm{HI}}_{\mathrm{UVA}}}{{\mathrm{HI}}_{\mathrm{UVC}}}\;\mathrm{and}\;{\mathrm{RDPC}}_{A,C}=\frac{{\mathrm{CI}}_{\mathrm{UVA}}}{{\mathrm{CI}}_{\mathrm{UVC}}}$).

RDP_A,C_	Line	
	OH	C=O
24 h	1.7	1.4
48 h	3.2	4.1

**Table 3 polymers-15-04322-t003:** Comparison of the effect of UV-A and UV-C on the formation of hydroxyl groups (OH) in PET Fiber and line samples in the bio-solids environment (${\mathrm{RDPH}}_{A,C}=\frac{{\mathrm{HI}}_{\mathrm{UVA}}}{{\mathrm{HI}}_{\mathrm{UVC}}}$).

RDP_A,C_	Fibre	Line
	OH	OH
24 h	~1.0	~1.0
48 h	1.9	1.1

**Table 4 polymers-15-04322-t004:** Comparison of hydroxyl and carbonyl indexes in polypropylene (PP) and polyethylene (PE) samples in bio-solid suspensions following 48 h of UV-A irradiation (this study) with the PP and PE samples in incubation for 36 months in natural conditions [41].

Type	PP Index Our Work (48 h)/PP Index Brandon’s Work (36 Months)			PE Index Our Work (48 h)/PE Index Brandon’s Work (36 Months)	
	Fiber	Line	Granule	Line	Granule
OH	4.2	1.7	1.2	21.7	1.0
C=O	5.2	2.9	2.2	18.8	1.8

## Data Availability

The data presented in this study are available on request from the corresponding author.

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
