# Peer review of "UV Light Causes Structural Changes in Microplastics Exposed in Bio-Solids"

_polymers, 2023, doi:10.3390/polym15214322_

Round 1
Reviewer 1 Report
Comments and Suggestions for Authors
I recommend the publication after minor revision:
1. In addition to thickness, can the authors provide more details to quantify fiber, lines and granules? Like diameter and length?
2. Can the authors elaborate on “The limited effect of UV light on PE and PET indicates the relative strength of these polymers compared to PP”? or provide some speculations on the different performances among different polymers?
Author Response
Dear Reviewer,
On behalf of ourselves and our co-authors, we thank you very much for giving us an opportunity to revise our manuscript, we appreciate the editor and reviewers very much for their work, positive and constructive comments, and suggestions on our manuscript entitled “UV light causes structural changes in microplastics exposed in bio-solids” (ID: 2622523).
We have studied the reviewer’s comments carefully and have made revisions which are marked in the paper. Both of the reviewer's comments are very welcome, as they improve the understanding and quality of the article. We have done our best and revised our manuscript according to the reviewer's comments. In the newly submitted article please find the revised version, which we would like to submit for your kind consideration, additionally below we added some explanation.
Reply to reviewer comments:
1. In addition to thickness, can the authors provide more details to quantify fiber, lines and granules? Like diameter and length?
Answer: Thank you for your comment. We added the missing information about other dimensions of fibers, lines and granules in the section Materials and methods, as follows:
Three types of polymers (polyethylene (PE), polypropylene (PP) and polyethylene terephthalate (PET), were tested in three different shapes: fibres, lines and granules. Granules can be named also pellets due to their cylindrical shape in the size of about 3 mm in diameter and about 5 mm in height – see Fig. S1. The thickness of fibres and lines was about 50 µm and 500 µm, respectively, while the length of lines was approximately 15 mm. Both were obtained from consumer factories.
2. Can the authors elaborate on “The limited effect of UV light on PE and PET indicates the relative strength of these polymers compared to PP”? or provide some speculations on the different performances among different polymers?
Answer: With the reviewer, we agree that the first sentence in section 4.3.: "The limited effect of UV light on PE and PET indicates the relative strength of these polymers compared to PP" is vaguely written and also without a reference that would adequately support it. Therefore, we thank reviewer 1 and we added the transformed sentence in section 4.3, to which we have added also 2 additional references, as follows:.
The results confirmed that ultraviolet irradiation affects the aging process of the tested polymers polyethylene (PE), polyethylene terephthalate (PET) as well as polypropylene (PP) [72]. They found a greater impact of UV light on PE and PET, with PE aging the easiest under artificial solar radiation [73].
- (Yun Kyung Lee, Kathleen R. Murphy, Jin Hur. Fluorescence Signatures of Dissolved Organic Matter Leached from Microplastics: Polymers and Additives. Environ. Sci. Technol. 2020, 54, 19, 11905–11914, https://doi.org/10.1021/acs.est.0c00942.
- Zhe Li, Yuwei Xie, Yi Zeng, Zihang Zhang, Yingyue Song, Zhicheng Hong, Lanqianya Ma, Mei He, Hua Ma, Fuyi Cui. Plastic leachates lead to long-term toxicity in fungi and promote biodegradation of heterocyclic dye, Science of The Total Environment, Volume 806, Part 1, 2022, 150538, https://doi.org/10.1016/j.scitotenv.2021.150538.
We would like to express our great appreciation to you and the reviewers for your comments on our paper.
Looking forward to hearing from you.

Reviewer 2 Report
Comments and Suggestions for Authors
The manuscript interestingly discusses the effect of degradation of microplastics present in the bio-solids under UVA and UVC irradiation at 70°C. The authors have monitored the degradation of microplastics through the ATR-IR technique by monitoring the changes in the hydroxyl and carbonyl groups before and after the microplastic's degradation. Also, the authors monitored structural changes after the degradation through SEM. However, the authors should address the following concerns before the acceptance of the publication.
1. The authors should mention the intensity of UVA and UVC radiation falling onto the sample in the materials and methods section.
2. Gaussian or Lorentzian fitting profiles should be used to calculate the CI or HI for all the samples. The calculation will be more accurate if the fitted profiles are used for the calculation.
3. The authors should calculate the carbonyl index/hydroxyl relative to the reference peak from the same spectra, i.e., for example, for 24 h UVA irradiation PP sample, CI should be calculated from maximum carbonyl peak adsorption/reference peak, instead of considering one reference peak of UVC for all the samples.
4. It’s better to mention the CI and HI of the sample without any irradiation (0 min).
5. The authors should try to give explanations also in the abstract, as to why the fibers are degraded more than the other types of plastics.
Comments on the Quality of English LanguageOnly minor editing of the English language is required.
Author Response
Dear Reviewer,
On behalf of ourselves and our co-authors, we thank you very much for giving us an opportunity to revise our manuscript, we appreciate the editor and reviewers very much for their work, positive and constructive comments, and suggestions on our manuscript entitled “UV light causes structural changes in microplastics exposed in bio-solids” (ID: 2622523).
We have studied the reviewer’s comments carefully and have made revisions which are marked in the paper. All of the reviewer's comments are very welcome, as they improve the understanding and quality of the article and/or point out to the authors areas where they could improve, which is also very much appreciated. We reviewed the reviewer's comments in detail and made every effort to revise our manuscript according to the reviewer's comments where possible. In the newly submitted article please find the revised version, which we would like to submit for your kind consideration, additionally below we added some explanation.
Reply to reviewer comments:
1. The authors should mention the intensity of UVA and UVC radiation falling onto the sample in the materials and methods section.
Answer: Thank you for your comment. We added the following sentence in the Materials and Methods section: The average UV irradiation in the chamber during the test was achieved with 200 W/ 0.36 m2, which means that 5 x 40 W lamps irradiated an area of 0.36 m2 (chamber 120 cm x 30 cm ), at individual wavelengths of 340 nm or 254 nm, respectively.
Exposure times and the lamp position (distance to samples) are already in the paper.
Additional explanation:
The reviewer states that the intensity of UVA and UVC radiation should be reported in the Materials and Methods section. Information about UV radiation is important in (accelerated) irradiation, so we agree with the reviewer that it is good to have this information for a better understanding of the article.
As we already mentioned in the article, we had 5 UV lamps and UV intensity sensors in the test chamber, as well as a control and recording unit, but due to the high UV light flux, all UV records were out of range of the sensor. Thus, we can only report that we had an average of 200 W/0.36 m2 at individual wavelengths of 340 nm and 254 nm, respectively. UV intensity is calculated and reported based on lamp wattage, number of lamps, and chamber dimensions.
As suggested by a reviewer, the average radiation intensity during the test, calculated from the data available to us, is now reported in the Materials and Methods section.
2. Gaussian or Lorentzian fitting profiles should be used to calculate the CI or HI for all the samples. The calculation will be more accurate if the fitted profiles are used for the calculation.
Answer: The opinion of the respected referee is correct, our calculations are without fittings and were based on the articles that were used as references in these articles. Additionally, to fit these results to experimental spectral data, software (e.g. Python) is usually used, which we do not have and have not used. This time we can't redefine this index but we agree with the reviewer and greatly appreciate the comment given, as it will increase the quality of our research work (maybe with new software) in the future as well. Thank you.
3. The authors should calculate the carbonyl index/hydroxyl relative to the reference peak from the same spectra, i.e., for example, for 24 h UVA irradiation PP sample, CI should be calculated from maximum carbonyl peak adsorption/reference peak, instead of considering one reference peak of UVC for all the samples.
Answer: Two approaches are possible. When using a single selected reference peak (often at a specific wavenumber) for all samples, the absorbance of that reference peak is used to normalize the absorbance of the carbonyl (or hydroxyl) peaks in each sample. The carbonyl index is then calculated as the ratio of the absorbance of the carbonyl peak to the absorbance of the reference peak. This allows relative comparisons between different samples. Alternatively, when using a reference peak from the same spectrum, the reference peak is present in the same spectrum as the carbonyl peak (or hydroxyl). This means that we select a vertex (usually unaffected by the change) as a reference. This approach is often preferred when you want to account for potential differences in sample preparation, instrument conditions, or other factors that might affect the absolute intensity of the peaks in the spectrum. From previous works, we found that the use of one reference peak for all samples simplifies the analysis and enables the comparison of different samples. However, it assumes that the reference peak remains constant across all samples, so readers should think critically as this is not always true. Using a reference peak from the same spectrum is more reliable in terms of accounting for variability between spectra and samples. It ensures that the reference and carbonyl peaks are measured under the same conditions. However, it may require more care in selecting the appropriate reference peak in each spectrum and may not allow direct comparison between different samples. In our research, we chose the first method based on the goal of comparing different samples. The stability of the reference peak, the importance of comparing different samples, and the various scientific works from this field to which we referred, decided on the approach we used.
4. It’s better to mention the CI and HI of the sample without any irradiation (0 min).
Answer: The CI and HI for all initial (labelled virgin) samples without irradiation (0 min) are given in the Supplement in graphical form.
5. The authors should try to give explanations also in the abstract, as to why the fibers are degraded more than the other types of plastics.
Answer: In the Abstract, we reformulated the sentence "Fibers were more degraded than the other two plastic shapes." into "Due to the surface-to-volume ratio of tested samples, which influences the degradation rate, fibers were more degraded than the other two plastic shapes.", with this we tried to give the explanation, also in the Abstract, why the fibers are degraded more than the other types of plastics.
Additional explanation:
A partial explanation of why fibers degrade more than other types of plastics is already given in section 4.3, where it is written: "Also for PE, the importance of thickness in creating oxidation groups was shown, with carbonyl and hydroxyl groups observed only in lines but not in granules. Also for PET samples, UV exposure only affected the line and fiber samples in the bio-solids suspensions, leading to an increased abundance of hydroxyl groups, with the fibers being most affected. This could again be due to their lower thickness and higher surface-to-volume ratio."
In section 2.2. UV and temperature exposures is a description of all three tested polymers, including their shapes and dimensions, where it follows that fibers are 20 microns, lines are 500 microns thick, and granules are 5 mm. Additionally, the authors provided more details to quantify fiber, lines, and granules.
Also, it was further noted in the conclusion: "Due to their highest surface-to-volume ratio, fibers showed the highest degradation, followed by lines and granules."

Round 2
Reviewer 1 Report
Comments and Suggestions for Authors
Recommend for publication
Reviewer 2 Report
Comments and Suggestions for Authors
The authors have provided sufficient explanation and improved the manuscript. I recommend the author insert CI and HI values for virgin samples in the supplementary information and give a short explanation in the main text of the manuscript. Currently, only spectra of the virgin samples are inserted in the supplementary information. After, this correction manuscript could be accepted for publication.
Comments on the Quality of English LanguageEnglish language is acceptable.
Author Response
Dear Editor/ Dear Reviewer.
Thank you very much for revisiting the article titled "UV light induces structural changes in microplastics exposed in biosolids" (ID: 2622523) and considering our responses. We reviewed the reviewer's comment and correct our manuscript according to the reviewer's comment. In the newly submitted article please find the revised version, additionally below we added some explanation.
Reply to reviewer comments:
“The authors have provided sufficient explanation and improved the manuscript. I recommend the author insert CI and HI values for virgin samples in the supplementary information and give a short explanation in the main text of the manuscript. Currently, only spectra of the virgin samples are inserted in the supplementary information. After, this correction manuscript could be accepted for publication.”
Based on the reviewer's last comment, we have added a sentence to the main text of the article noting the importance of initial (virgin) samples: "The specific peaks used as a reference and the exact calculation may vary depending on the material used (sample source, age, storage, preparation, and other effects on the sample). For this reason, the CI and HI were calculated for virgin samples (the samples at 0 min of UV irradiation). These values are around 1 and presented in Table S1 in the Supplementary materials."
Data on the CI-index and HI-index of the virgin samples were added in the Supplementary section as suggested by the reviewer. In the new, updated version of the article, the specified data is submitted for final consideration.
By complying with all the comments of all the reviewers, the understanding and quality of the article are better. Therefore, we thank the reviewers for their work, positive and constructive comments and suggestions on our manuscript, and the editor for their support.
Yours sincerely,
Branka Mušič and Andrijana Sever Škapin

Round 3
Reviewer 2 Report
Comments and Suggestions for Authors
The authors have sufficiently revised the manuscript and hence it could be considered for publication.